# Voltammetry in Determination of Trace Amounts of Lanthanides—A Review

**DOI:** 10.3390/molecules28237755

**Published:** 2023-11-24

**Authors:** Malgorzata Grabarczyk, Marzena Fialek, Edyta Wlazlowska

**Affiliations:** Department of Analytical Chemistry, Institute of Chemical Sciences, Faculty of Chemistry, Maria Curie-Sklodowska University, 20-031 Lublin, Poland; malgorzata.grabarczyk@mail.umcs.pl (M.G.); edyta.wlazlowska@onet.pl (E.W.)

**Keywords:** review, rare earth elements, stripping voltammetry method, working electrode, complexing agent, interferences, application

## Abstract

This paper presents an overview of approaches proposed in the scientific literature for the voltammetric determination of rare earth elements (mainly cerium and europium individually, as well as various lanthanides simultaneously) in manifold kinds of samples. The work is divided into chapters describing the most important aspects affecting the sensitivity of the proposed methods: the technique adopted (AdSV, ASV, CSV), complexing agents used, the kind of working electrode (mercury-based, noble metal or carbon electrodes) and the most popular electrode modifiers (e.g., metal film, carbon nanotubes, molecularly imprinted polymers). Analytical parameters of the procedures presented in the paper are collected in tables. The subsequent chapters are devoted to a detailed discussion of potential inorganic and organic interfering factors. The possibilities of simultaneous determination of several lanthanides in one sample and the influence of other lanthanides on the determined rare earth element were also discussed. Finally, the applications of the voltammetric procedures to the determination of rare earth metals in real samples with miscellaneous matrix is described. All analytical results were tabulated in order to compare the analytical suitability of the proposed procedures.

## 1. Introduction

In accordance with the enjoinder by the International Union of Pure and Applied Chemistry (IUPAC), rare earth elements (REEs) are a group of chemical elements that includes the group of lanthanides (lanthanum, cerium, praseodymium, neodymium, promethium, samarium, europium, gadolinium, terbium, dysprosium, holmium, erbium, thulium, ytterbium, lutetium), with atomic numbers from 57 to 71, as well as scandium and yttrium [1]. These metals have similar physical and chemical properties. The elements considered tend to create basic oxides (with the exception of cerium(IV) oxide, which is amphoteric) in which they occur mainly in the +III oxidation state, but sometimes also in +II or +IV. Of all of them, only promethium is a synthetic radioactive element. Lanthanides are divided into two main groups. La, Ce, Pr, Nd, and Sm are among the light lanthanides, while Eu, Gd, Tb, Dy, Ho, Er, Tm, Yb, and Lu are the so-called heavy lanthanides [1,2,3,4].

More than 95% of the rare earth elements in the Earth’s crust are light lanthanides. The contribution of the other heavy REEs is less than 5% altogether, and truly in this case their common name suits them superbly. It is also worthwhile noting that, in the main, rare earth elements are never found in a free state, but they always co-occur in diverse minerals. In nature, they predominantly occur in the form of bastnaesite (Ce,La)FCO_3_, monazite (Ce,La,Y)PO_4_, and xenotime (YPO_4_). The geological resources of lanthanides are significant, but their concentration in mined ore is low, in the order of fractions of a percent [5,6].

Currently, rare earth metals are used in the most dynamically developing areas of the high technology industry, such as aviation, space flights, production of mobile phones (smartphones), catalysts, high-energy magnetic materials, LCD screens, LED diodes, hybrid car engines, and new generation Ni-MH batteries. These metals are widely used in metallurgy as alloying additives to improve the properties of doped metals, permanent magnets or polishing pastes. The biological activity of lanthanum compounds has also been proven and, hence, they are used in medicine. They have also found a unique application in the production of optical filters, phosphors, dyes, fertilizers, and insulation fibers [2,4]. REEs have become indispensable in the world of technology, owing to their unusual magnetic, phosphorescent, and catalytic properties [7]. The growing demand for these elements has resulted in these metals being included in the group of 20 critical mineral raw materials for the EU economy [2].

The main environmental risk posed by rare earth elements is tailings, which are a mixture of small-sized particles, waste water, and floatation chemicals used in the processing stages [8]. Most rare earth elements also consist of radioactive materials which impose the risk of radioactive dust and water emissions [9]. This is especially dangerous given the fact that REEs can accumulate in the human body and lead to a number of diseases such as leukemia, acute myocardial infarction, abnormal blood biochemical parameters, as well as skin lesions [10,11]. The abundance of electrical gadgets around us and agricultural and industrial use of rare earth elements also pose environmental hazards if not properly monitored [12]. Therefore, there is an increasing need to determine the concentration of REEs in the environment in order to evaluate the potential threat to the aquatic ecosystem.

The similarity of rare earth elements makes their determination remarkably tough and complicated. Especially significant issues emerge if the selected rare earth metal must be determined in a mixture with the other REEs due to a lot of possible disruptions. Over the years, it can be seen that the interest of scientists in the determination of rare earth metals by various methods is growing. So far, the instrumental techniques used for the detection and quantification of lanthanides include spectroscopic techniques such as graphite furnace atomic absorption spectrometry (GFAAS) [13], inductively coupled plasma optical emission spectrometry (ICP-OES) [14,15,16] and X-ray fluorescence spectroscopy (XRF) [17,18]; spectrometric technique, i.e., inductively coupled plasma mass spectroscopy (ICP-MS) [19,20,21] or spectrophotometry [22]; and other techniques such as neutron activation analysis (NAA) [23] and high-performance liquid chromatography (HPLC) [24]. NAA is a very sensitive approach, but the chief downsides of this method are the need for a source of neutrons as well as the irradiated samples becoming radioactive. Therefore, strict safety precautions must be adopted to avoid risks for the operators and the environment. As far as ICP-OES is concerned, solid samples cannot be analyzed, but this technique is repeatedly used in the determination of REEs because of its capability for rapid multi-element detection over a wide concentration range. However, the concentration of REEs in samples is usually much lower than the limits of detection that can be achieved by this technique, and the major constituents, namely organic compounds and inorganic salts, can be the cause of matrix effects. The performance of the instruments in XRF is poor for REEs because of the intricacy of the emission spectrum as well as the amount of interference from the major elements. Therefore, the detachment of the matrix through cation exchange is usually required. HPLC in conjunction with various detectors (for example, electrospray mass spectrometry (ESI-MS)) offers fast and efficient separation as well as quantification of individual lanthanides, but in the process, this method is intricate and requires costly equipment. The above-mentioned techniques, despite their high sensitivity, often do not provide sufficient selectivity and unfortunately involve complex and costly maintenance and operation [25].

The electrochemical approach to the determination of trace lanthanides, such as potentiometry [26,27,28] and voltammetry [29,30,31,32,33,34,35,36,37,38,39,40,41,42,43,44,45,46,47,48,49,50,51,52,53,54,55,56,57], can compete with the above-mentioned methods due to their ease of handling, great sensitivity, quick response, as well as cost efficacy. At the same time, these techniques are characterized by a detection limit comparable to the aforementioned spectroscopic techniques, but the most sensitive and effective electrochemical technique for trace measurement of many analytes is stripping voltammetry. Stripping analysis is one of the most sensitive electroanalytical techniques because of the embodiment of an electrolytic preconcentration step before the actual electrode process, which leads to extremely low detection limits. A two-stage analytical measurement allows obtaining a limit of detection at the ppb and even sub-ppb level. Depending on whether the preconcentration process is taking place as a result of electrolysis, adsorption or electrode reaction leading to the formation of a sparingly soluble compound on the electrode surface, stripping analysis is divided into the following techniques: anodic stripping voltammetry (ASV), adsorption stripping voltammetry (AdSV), and cathode stripping voltammetry (CSV). Adsorptive stripping voltammetry is known to be useful for many analytes that cannot be accumulated electrolytically [58]. 

The review of the literature showed that among the methods for voltammetric determination of REEs, most of them were developed using AdSV [29,30,31,32,33,34,35,36,37,38,39,40,41,42,43,44,45,46,47,48,49,50,51,52,53,54,55]. On the other hand, in database resources, it is difficult to find works focused on the direct determination of these elements using the ASV method. Only one paper concerning the indirect method of lutetium determination using the ASV method was found [56]. One work was also found regarding the determination of cerium using the CSV method [57]. This is because rare earth cations are special compounds for which effective preconcentration can only be achieved by using non-electrolytic preconcentration techniques [58]. Though they can be reduced to metals, which are to a certain extent soluble in mercury, conventional anodic stripping measurements are not feasible because of the poorly defined redox reactions that occur at extremely negative potentials [59]. The REE(III) ions are reduced in the potential range from −1.7 to −2.0 V and their peak currents are obtained at the potentials close to the potentials of the supporting electrolyte reduction and therefore are poorly selective. On the other hand, lanthanides have a stable +3 oxidation state that induces chelating agents to be able to form stable neutral complexes [60]. Therefore, ions of lanthanides are most often detected by adsorptive stripping voltammetry based on the signal from these elements’ complexes with an appropriate ligand.

A search of the literature on the voltammetric determination of lanthanides reveals that most of it is devoted to the voltammetric determination of cerium [29,30,31,32,33,34,35,36,37,38,39] and europium [40,41,42,43,44,45,46,47]. These metals are considered the most reactive rare earth elements. It is possible to determine both cerium and europium independently in the presence of other rare earths. Europium has the most stable +2 oxidation state of all the lanthanides and this feature affects its reduction at the electrode and causes the Eu(III) signal to be separated from the signals of other lanthanides [47]. As for cerium, its content in mineral and metallurgical samples is generally much higher than that of other rare earths and therefore it is possible to determine cerium in the presence of other rare earths without prior separation [31].

In the present work, a review of methods for the voltammetric determination of rare earth metals described in the literature is carried out. For this purpose, the most relevant analytical parameters of all procedures reviewed, such as limit of detection, linear range, accumulation time, the type of working electrode, as well as the applied complexing agent, have been collated in Table 1. The table also lists the tested potential interferents. The focus is on a detailed discussion on the topic of working electrodes and complexing agents used in the voltammetric methods reviewed. The procedures are also discussed in terms of the investigated interferences and their practical application.

## 2. Stripping Voltammetry Methods for the Determination of Rare Earth Elements

### 2.1. AdSV Procedures of REEs Determination

As already mentioned in the introduction to this work, the AdSV method has been widely used for the voltammetric determination of different rare earth elements. To be able to determine elements using this method, they must form stable complexes with an appropriately selected complexing agent. In the form of complexes, these elements adsorb on the electrode surface without undergoing any electrolysis processes. Therefore, the complexing agent used has a huge impact on the signal of the determined elements, and thus on the sensitivity of voltammetric analysis. The complexing agents applied in the voltammetric procedures of REE quantification, including the detection limits achieved in these methods, are presented in Table 1. As can be seen, the most frequently selected complexing agent of REEs is Alizarin [29,30,47,51,52]. The following ligands are used less often in adsorptive stripping analysis of lanthanides: Alizarin S [31,53], cupferron [40], mordant red 19 (MR19) [54], mixed complex of 2-thenoyltrifluoroacetone (TTA) and polyethyleneglycol (PAG) [55], o-cresolphthalexon (OCP) [49], and solochrome violet RS (SVRS) [56]. In each of the above procedures, the complexing agent was added directly to the tested sample during the analysis. As regards the other procedures, the role of the complexing agent was played by a modifier that was a component of the working electrode and there was no need to add any ligands to the solution.

When reviewing the data obtained during the optimization of the Alizarin concentration in the works [29,30,47,51,52], it can be concluded that each of these works shows that an increase in the Alizarin concentration causes an increase in the peak current of the tested REE up to a certain point and then the signal value stabilizes at a constant level, while a further increase in the concentration of the complexing agent causes a decrease in the signal value. The introduction of higher concentrations of Alizarin into the solution causes a decrease in the voltammetric response, probably due to the adsorption of the free ligand instead of the formed complex on the electrode surface, resulting in the blocking of the active sites of the electrode [51,52]. In the mentioned works, REE determinations were carried out at a pH equal to or close to 5.0. In the methods in which chemically modified electrodes such as the CTAB/CPE and GC/SbFE were used, the lowest Alizarin concentration of 2 × 10^−6^ M was applied as optimal [30,48]. However, in the procedures based on the use of an unmodified carbon paste electrode, the determination of REEs was carried out at a higher concentration of Alizarin equal to 3.2 × 10^−6^ M [29] and 4 × 10^−6^ M [51,52], respectively. A similar relationship between the concentration of the complexing ligand and the REE signal was also observed in the case of Alizarin S [31,53] and cupferron [40]. However, in the work [31], due to an increase in Alizarin S concentration, a sharp increase in the Ce(III) signal is observed; subsequently, the maximum value of the signal is reached at a concentration of 3 × 10^−5^ M, and then the peak slowly decreases without a plateau. 

It can be noted that depending on the developed procedure, the use of Alizarin as a complexing agent enables the determination of light or heavy rare earth elements. For example, the procedure [51] developed using Alizarin is more sensitive to heavy rare earths (Dy, Ho, Er, Tm, Yb, and Lu) than to light ones. The procedure [52] is dedicated to the determination of scandium only, whilst both methods [29,30] are suitable for the detection of cerium. In all the above-mentioned procedures, the CPE was used as the working electrode. However, in the work [48], the GCE/SBF sensor gave the best voltammetric response to cerium, whilst the responses to lanthanum and praseodymium were lower. Because of a significant variance between the peak potentials of Ce(III)-Alizarin and other rare earth(III)-Alizarin complexes, Alizarin is the most often used to determine cerium individually in the presence of other rare earths. The use of mordant red 19 (MR19) as a complexing agent in the paper [54] allowed indirect analysis of both light and heavy lanthanide ions and even simultaneous determination in certain lanthanide mixtures. Despite the similar chemical properties of all lanthanides, the MRl9 complex system used in this work can differentiate between light lanthanide ions and heavy ones because the variance in peak potentials between the lanthanide-MR19 complex and free MR19 increases with the increase in the atomic number of the lanthanide ion. To be exact, using MR19 as a complexing agent, separate signals individually for each lanthanide can be obtained. Although the smallest potential difference occurs for La(III), it is large enough to differentiate the complex peak from the signal coming from the free MR19 [54]. Nevertheless, heavy rare earth elements cannot be determined using a complexing ligand, o-cresolphthalexon, which was applied by Wang et al. to simultaneously determine lanthanum, cerium, and praseodymium [49]. This is related to the decreasing potential variance between the complex and the free ligand as the atomic number of the lanthanide increases. On the other hand, the use of Solochrome Violet RS [56] made it possible to determine only heavy rare earths. To sum up, OCP is sensitive only to light rare earth elements [49], SVRS is sensitive to heavy rare earth elements [56], while MR19 shows a similar detection level for both light and heavy rare earth ions. Due to the similarity of the peak potentials of complexes formed between Alizarin/MR19/OCP/SVRS and different REEs (see Table 1), it is necessary to separate the lanthanide ion mixtures before the voltammetric analysis [48,49,51,54,56].

Mlakar described the procedure of Yb(III) quantification in which he used the mixed ligand system TTA-PEG forming stable complexes with Yb(III) strongly absorbing on the electrode surface in ammonium chloride. Additionally, NH4Cl has catalytic properties, which increases the sensitivity of the method [55].

In contrast to the above-mentioned procedures, other AdSV methods of lanthanide determination are based on the use of modifiers of solid electrodes which enable performing a non-electrolytic lanthanide preconcentration stage. In these procedures, there is no need to additionally introduce complex agents into the tested solution. Modifiers acting as complexing agents are as follows: dipyridyl (DP) [32], N′-[(2-hydroxyphenyl)methylidene]-2-furohydrazide (NHMF) [33], but-2-enedioic acid bis-[(2-amino-ethyl)-amide] [34], vinylpyridine (VP) and methacrylic acid (MA) [35,41], allyl phenoxyacetate (APA) [36], poly(catechol) (PC) [37,38,42], o-phenylenediamine (OPD) [38], 3-methyl-2-hydrazinobenzothiazole (MBTH) [50], 2-pyridinol-1-oxide (PO) [43] and salicylamide (SA) [44]. Procedures in which the above modifiers were used are discussed in more detail in the chapter devoted to working electrodes and the modifiers used.

On the other hand, no complexing agent was used in the AdSV procedure described in the works [45,46,47]. Instead of this, ion-exchange preconcentration of both europium and iterbium on Nafion-coated thin mercury film electrodes (NCTMFE) was proposed in the work [46]. However, in the method [45], as a result of the exchange of Eu(III) by Nafion and subsequent electrostatic adsorption on the surface of the multiwall carbon nanotubes film (MWCNTs), the maximum Eu(III) incorporation into the composite film is achieved. In the work [47], the electrostatic adsorption of Eu(III) on a monolayer of a surfactant such as sodium dodecylbenzene sulfonate (SDBS), formed on the surface of the LaB6 electrode, was also reported.

### 2.2. ASV Procedure for Lutetium(III) Determination

Kumric et al. described an indirect anodic stripping voltammetry procedure for Lu(III) quantification, based on the substitution reaction between Lu(III) and Zn-EDTA. Since the reduction potential of Lu(III) at a mercury electrode is greatly negative, close to the decomposition potential of the supporting electrolyte, direct determination of Lu(III) by the ASV method is impossible. However, this study took advantage of the fact that Zn forms less stable complexes with EDTA than Lu(III) and gives a well-developed voltammetric peak. In this procedure, therefore, upon adding lutetium(III) to the solution, an equivalent amount of Zn(II) is released from the complex into the solution, which can be easily measured by the ASV method. The signal obtained for zinc released from the complexes is directly proportional to the amount of lutetium in the sample. In addition, due to the fact that the stability constants of lanthanide complexes increase with increasing atomic number, lutetium can be determined in the presence of other lanthanides without the need for tedious separation. Using this method, lutetium can be determined in the concentration range from 2.1 × 10^−9^ to 7.3 × 10^−6^ M [57].

### 2.3. CSV Procedure for Cerium(III) Determination

Ojo et al. developed a cathodic stripping voltammetry procedure for cerium determination in which accumulation of cerium(III) as insoluble CeO_2_ was carried out on the surface of the Indium tin oxide (ITO) electrode. In the stripping step, as a result of reduction, the cerium oxide formed was removed from the electrode surface and Ce(III) was released back into the solution. In this procedure, to achieve a lower detection limit, Osteryoung square wave voltammetry (OSWV) was used for the stripping step due to its capability to minimize non-faradaic current. A well-defined peak of Ce(III) was obtained in the potential area with a very smooth background current and therefore it can quantified with no problem, which is especially important in the CSV method. This CSV method is suitable for determining cerium concentrations in the range of 1 × 10^−7^ to 7 × 10^−7^ M, with a detection limit equal to 5.8 × 10^−9^ M [39]. 

### 2.4. Types of Working Electrodes and Electrode Modifiers Used

#### 2.4.1. Mercury-Based Electrodes

Up to 2000, voltammetric methods were most often based on the use of mercury electrodes characterized by excellent adsorption properties, ideal polarizability, smooth surface, good signal repeatability, and a wide range of negative potential values. Unfortunately, due to the ease of metal oxidation, mercury electrodes have a relatively limited application in the anode area. In voltammetric determination of trace amounts of rare earth elements, mercury electrodes, such as the hanging mercury drop electrode (HMDE) [40,55,56,57], the static mercury drop electrode (SMDE) [49,54], and the Nafion-coated thin mercury film electrode (NCTMFE) [46], have been used. Both the HMDE and SMDE are mercury electrodes with a special design that enables the voltammetric process to be carried out on one drop of mercury. A great convenience in using mercury drop electrodes is that there is no need to specially prepare their surface, and what is more, it is renewed periodically so no contamination accumulates on them. Nevertheless, this electrode structure does not allow for a large stirring speed for fear of detaching the drops, and the peaks in the voltammogram recorded using it are slightly wider and lower than those obtained using a mercury film electrode (the time of diffusive metal transport during electrolytic dissolution is longer in the case of drops than in the case of a thin film). However, the use of mercury electrodes leads to the release of toxic mercury into the environment. The growing risks associated with the use, handling, and disposal of metallic mercury and its salts required for mercury film manufacture restrict the usage of mercury in laboratory practice. Therefore, there is an attempt to search for new materials that would allow obtaining electrodes that would have the advantages of mercury electrodes and at the same time would be less toxic.

The first work on the determination of lanthanides using the AdSV method was published in 1985. Wang et al. attempted to determine light lanthanides such as lanthanum, cerium, and praseodymium using an SMDE electrode [49]. A slightly more frequently selected electrode for the determination of REEs has been the HMDE. Using this electrode, the AdSV procedures for Eu(III) [40], Yb(III) [55], Lu(III) [57] as well as Y(IIII), Dy(III), Ho(III), and Yb(III) [56] determination have been developed. The lowest detection limit of 1.6 × 10^−11^ M was obtained in the procedure of Eu(III) determination based on the complexation of Eu(III) with cupferron. Another mercury electrode used to determine Eu(III) and Yb(III) was a Nafion-coated thin mercury film electrode (NCTMFE) [46].

#### 2.4.2. Solid Electrodes

An alternative to mercury electrodes is solid electrodes. Frequently used solid electrodes are those made of noble metals (Pt, Au, Ag) or carbon electrodes (glassy carbon, graphite, paste electrode). Unlike mercury electrodes, the above-mentioned electrodes are non-toxic and can be used at both negative and positive potentials. Their use makes it possible to determine ions present in the sample as a result of their reduction or oxidation process. These electrodes are highly stable in various solvents. Additionally, they can be easily prepared as well as chemically modified. Among solid electrodes, only carbon paste electrodes are characterized by a renewable surface. For the other carbon electrodes, as well as the noble metal-based ones, the surface can be renewed only after time-consuming mechanical and electrochemical treatments.

Solid electrodes for the determination of REEs include carbon-based electrodes, such as glassy carbon electrodes (GCEs) [31,37,38,45,48], carbon paste electrodes (CPEs) [29,30,32,33,35,36,41,50,51,52,53] and to a lesser extent gold electrodes [43]. Looking through the literature, it can be safely stated that at the end of the 20th century both GC and CP electrodes, with chemical modifications, received growing importance in the analysis of trace elements, including lanthanides, particularly when used coupled with stripping analysis. The modifier, selected to have high propinquity for the analyte, provides increased selectivity combined with high sensitivity, emerging in the non-electrolytic pre-concentration stage prior to voltammetric analysis [61,62]. Glassy carbon electrodes (GCEs) are generally used as a substrate for film electrodes. In voltammetric determination of REEs, glassy carbon has been used as a substrate for poly(catechol) film [37], poly(catechol) and ion-imprinted membrane [38], antimony film [48], as well as multiwall carbon nanotubes and Nafion composite film [45]. On the other hand, carbon paste electrodes made of graphite grains mixed with a nonconductive oily organic liquid can be easily modified by incorporating a variety of ligands into the paste. Therefore, in several procedures of REE quantification working electrodes were based on a modified carbon electrode [30,32,33,35,36,41,50]. In the latter works, the carbon paste electrode was modified by using cetyltrimethylammonium bromide [30], dipyridyl-functionalized nanoporous silica gel [32], N′-[(2-hydroxyphenyl)methylidene]-2-furohydrazide (NHMF) [33], cerium-imprinted polymer and multiwalled carbon nanotubes [35], ion-imprinted polymers [36,41] and 3-Methyl-2-hydrazinobenzothiazole (MBTH) [50].

In accordance with the data in Table 1, the most sensitive (*LOD* = 1 × 10^−12^ M) and highly selective AdSV method for the determination of lanthanides was obtained in the work [50], in which MBTH was used as the CPE modifier. This organic ligand containing in its structure an N- and S-based complexing center is capable of selectively coordinating with transition and heavy metals [23]. As described in the article [50], it also forms stable complexes with lanthanum. Javanbakht et al. developed two AdSV procedures for cerium determination using a CPE electrode modified with organic ligands, such as dipyridyl (DP) [32] and NHMF [33], and which directly coordinated with cerium(III). DP forms a complex with Ce(III) through two nitrogen atoms of the pyridine ring, while NHMF coordinates with Ce(III) via donor oxygen and nitrogen atoms.

Both methods can be used to determine cerium concentrations in the range of 10^−9^–10^−8^ M, but a slightly lower detection limit of 8 × 10^−10^ M was obtained in the procedure [33]. On the other hand, in the paper [43] on determining europium, 2-pyridinol-1-oxide (PO) was used as a modifier of a gold electrode. The applied organic ligand forms coordination bonds with europium through two oxygen atoms. According to Table 1, this method is characterized by the highest detection limit (equal to 3 × 10^−7^ M) of all voltammetric methods dedicated to lanthanide quantification.

Recently, modified screen-printed electrodes have also been used for REE analysis [34,42,44]. The following materials have been used as modifiers: double-ion imprinted polymer @ magnetic nanoparticles [34], ion-imprinted membrane and poly(catechol) [42], and salicylamide self-assembled monolayers on mesoporous silica [44]. Screen-printed electrodes (SPEs) have currently drawn considerable attention due to several advantages of these sensors such as low cost, high repeatability of the obtained electrodes, flexibility of their design, the possibility of producing them from various materials, and wide possibilities of modification of the working surface. In addition, these electrodes can be connected to portable equipment enabling in situ quantification of specific analytes. Moreover, SPEs often do not need electrode pretreatment or electrodeposition and/or electrode polishing, dissimilar to other electrode materials [63].

Carbon nanotubes are a popular electrode material and they have been applied as a modifier for both CP [35] and GC electrodes [45] due to their distinctive electronic structure, electrical conductivity, large specific surface area, as well as strong adsorption capacity [64]. By using MWCNTs-Nafion film to modify a GCE, the sensitivity of Eu(III) determination was improved because of the catalytical action of MWCNTs and Nafion film’s capacity to accumulate cations. This procedure is characterized by a limit of detection equal to 1 × 10^−8^ M. Over the last few years, special attention has been attracted to ion-imprinted polymers (IIPs)/membrane (IIM) that have the ability to recognize specific lanthanide ions. Generally, ion-imprinted material is characterized by great selective appreciation, stability, reusability, simplicity, and low cost in preparation. As mentioned earlier, this modifier has been used in as many as six voltammetric methods of lanthanide detection [34,35,36,38,41,42]. The researchers have reported that the imprinted materials can most often be incorporated into carbon paste electrodes, which allows for the development of highly selective sensors for the determination of different kinds of molecules or ions. The development of imprinted materials for lanthanide ions is of particular importance due to the widely known issue of separation of lanthanide ions [65,66,67]. In 2015, scientists from the Banaras Hindu University in India managed to successfully prepare an electrochemical sensor for the simultaneous determination of two lanthanide ions, Ce(IV) and Gd(III), via a dual ion-imprinting approach. The proposed sensor was found to be highly selective and sensitive for the simultaneous quantification of Ce(IV) and Gd(III). The detection limits obtained were 5.0 × 10^−10^ M and 1.2 × 10^−9^ M for cerium and gadolinium, respectively [34]. The presence of both multiwalled carbon nanotubes and Ce(III) imprinted polymer nanoparticles in the carbon paste electrode composition allowed a group of researchers led by Alizabeth to obtain a sensor with a low detection limit equal to 1.0 × 10^−11^ M [35]. Applied in the work [37], a glassy carbon electrode modified with poly(catechol) (PC-GCE) shows a cerium detection limit of 2.0 × 10^−10^ M, whereas using a poly(catechol) film modified glassy carbon electrode (PC-GCE) additionally modified with Ce(III) ion-imprinted membrane (IIM), it is possible to obtain a much lower detection limit equal to 1.0 × 10^−12^ M [38]. It has been confirmed that due to the presence of ion-imprinted sites created in ion-imprinted polymers, the electrochemical active surface is larger, and, therefore, a larger amount of the analyte is able to adsorb on the electrode surface. As for procedures in which the working electrodes were modified with poly(catechol), a significant enhancement of the voltammetric response was noticed due to the fact that poly(catechol) forms coordination bonds with lanthanides [37,38,42].

By analyzing the data collected in Table 1, it can be concluded that the voltammetric methods using carbon-based electrodes chemically modified with two different modifiers are characterized by wide linearity ranges covering 4 [42,45], 5 [35] or even 8 orders of magnitude [38].

A novelty in the voltammetric determination of REEs has been the use of indium tin oxide (ITO) working electrode material for the determination of cerium by CSV. This electrode is particularly suitable for CSV determinations due to its excellent positive potential range as well as a smooth background current related to metal electrodes such as platinum and gold which have interfering oxide waves. Additionally, the use of an ITO electrode does not require a complicated surface modification process [39]. Another novelty electrode material is lanthanum hexaboride (LaB_6_), which was used to determine europium in combination with sodium dodecylbenzene sulfonate as an ionic surfactant [47].

## 3. Impact of Interferents on REE Determination

Stripping voltammetry methods, like any other analytical method, are not free from interference. In adsorptive stripping voltammetric methods and other voltammetric techniques, interference is mainly determined by two factors, namely competition at the surface with the organic matrix, which is an inherent part of every real sample, and competition for the complexing agent from other metal ions. In most cases, these factors negatively affect the signal of the determined ion, including rare earth ions, as discussed below.

### 3.1. Impact of Co-Existing Ions

Interference from co-existing ions most often occurs as a result of forming a complex with the ligand used to complex the determined REEs and adsorption of the formed complexes on the electrode surface instead of complexes with the REEs. In such a case, the registered voltammogram shows a decrease in the analytical signal in relation to the signal obtained without foreign ions or its complete disappearance. The REE signal has been measured in the presence of ions commonly found in real samples such as Ag(I), Al(III), As(III), As(V), Ba(II), Bi(III), Ca(II), Cd(II), Co(II), Cr(III), Cu(II), Fe(II), Fe(III), Hg(II), K(I), Mg(II), Mn(II), Mo(VI), Na(I), Ni(II), Pb(II), Sb(III), Se(IV), Sn(II), Sn(IV), Zn(II), SO_4_^2−^, PO_4_^3−^, CH_3_COO^−^, C_2_O_4_^−^, Br^−^, Cl^−^, F^−^, and ascorbic acid [29,30,31,32,33,34,35,36,37,38,39,40,41,43,45,46,47,48,49,50,51,52,53,55]. Moreover, most of the studies have not omitted to check whether the analytical signal is not disturbed by the presence of other lanthanides in the developed procedures [29,30,32,33,34,35,36,38,39,41,45,46,47,50,51,52,53,55].

#### 3.1.1. Impact of Co-Existing Ions on Cerium Signal in Cerium Determination Procedure

The AdSV procedure for cerium determination using a carbon paste electrode (CPE) modified with DPNSG is free from interferences from tested foreign ions, except for lanthanum. This element in amounts greater than three times excess in relation to cerium significantly interferes with the cerium signal, reducing it. This is related to the strong competition of lanthanum with cerium in forming a complex with dipyridyl [32]. During the Ce(III) quantification carried out on a CPE modified with NHMF, interferences from lanthanum and samarium were found, which was also due to the competition of these metals with cerium to form complexes with NHMF [33]. The disturbing nature of lanthanum is also observed in the work [48] devoted to the simultaneous determination of La(III), Ce(III), and Pr(III) with Alizarin as a complexing agent, but La(III) interferes more strongly with praseodymium than with cerium. By examining interference in the Yb(III)-TTA-PEG system, Mlakar et al. observed that both lanthanum and europium give distinct signals in the voltammogram, but the presence of these ions did not affect the height of the Yb(III) peak [55]. In the cerium–Alizarin system using a CPE [29] as well as a CTAB/CPE [30] as the working electrode, at least a 2-fold excess of all rare earth elements did not cause any interference. In the procedure [31] based on using a GCE and Alizarin as a complexing agent, it was shown that Cr(III), Fe(II), Sb(III), and V(V) are characterized by the most interfering nature, as a 50-fold excess of these interferents caused the signal of Ce(III) to be reduced by 70%, 50%, 65%, and 30%, respectively. The measurements carried out using a poly(catechol)-modified GC electrode indicate that a 50-fold excess of Bi(III) and a 20-fold excess of Ta(V) diminished the Ce(III) peak current by 38% and 25%, respectively, but no additional signals appeared for the foreign ions. These interferences were apparently caused by competitive accumulation on the polymer surface [37].

When investigating the selectivity of Ce(III)-IIM/PC/GCE towards Ce(III), it was shown that a 50-fold excess of Fe(III), Cu(II), and Ni(II) entails a reduction in the AdSV voltammetric response by about 18%, 8%, and 9%, respectively. But a 20 fold of other rare earth elements is acceptable. The results are depicted in Table 2 [38].

The work [35] reports that the voltammetric response of the Ce-IP/MWCNT/CP electrode to Ce(III) is not influenced by the presence of a 50-fold excess of La(III), Nd(III), Sm(III), Tb(III), and Yb(III). However, the same concentration of Eu(III) and Dy(III) causes significant disruption of the cerium signal. The Ce-IIPs-CP electrode developed by Chen et al. [36] has even better selectivity as it can be seen (Table 3) that the voltammetric response of this sensor to Ce(III) is not affected in the presence of a 500-fold excess of Er(III), Yb(III), Gd(III), Dy(III), Ho(III), Eu(III), Nd(III), Pr(III), and Tb(III). Such excellent selectivity allows it to be applied for direct quantification of cerium in a complex matrix. The high selectivity of the IIPs-CPE is due to specific sites occurring in IIP, which are only complementary to the template ions’ structure and can ensure strong coordination with the template ion [36]. 

The indium tin oxide (ITO) working electrode for the determination of cerium by the CSV method proposed by Ojo et al. shows satisfactory selectivity as most metals do not form insoluble oxides at higher oxidation states, which makes them insensitive to CSV. Mn(II), small amounts of which amplify the cerium signal, turned out to be the only significant interferer. This is because both Ce and Mn form insoluble oxides in higher oxidation states, which means that both CeO_2_ and MnO_2_ are removed from the ITO electrode at the same reduction potential [39].

In summary, all procedures for the individual determination of cerium have been tested in terms of selectivity. However, potential interferences of other lanthanides have not been checked in the works [31,37], whilst, in the paper [36], no attempt to investigate the influence of other foreign ions on the cerium signal was made.

#### 3.1.2. Impact of Co-Existing Ions on Europium Signal in Europium Determination Procedures

The experimental data point out that the AdSV response of Eu(III) to the N/MWCNTs/GCE is influenced by other REEs, such as Yb(III), Sm(III), Er(III), and La(III). Even though the tested ions do not give a voltammetric response, an increase in their concentration in the tested sample causes a decrease in the europium signal, apparently due to the similar structure of these cations to Eu^3+^ and the strong capacity to replace europium in the Nafion layer or the irreversible adsorption properties on MWCNT. Hence, due to competition for the migration and active sites on the modified electrode surface, the Eu(III) signal is reduced as a result [45]. Similarly, in the work [46] competition between La(III) and analytes for the ion exchange sites of the Nafion layer was reported. Ugo et al. showed that quantification of Eu(III) and Yb(III) on the Nafion-coated thin mercury film electrode (NCTMFE) in the presence of a 50-fold excess of La(III) as an interfering factor results in a decrease in sensitivity as well as an increase in the detection limit of Eu(III) and Yb(III). However, in [46], there is no data on the influence of other lanthanides on the analytical signals of europium and ytterbium.

In the work [40], concerning the determination of Eu(III) in the form of a complex with cupferron on an HMDE electrode, a ten-fold excess of single metal ions in relation to the concentration of europium, such as aluminum, chromium, molybdenum, uranium and vanadium which are known to form complexes with cupferron, was examined. It was found that chromium did not cause any interference, while after adding both molybdenum and vanadium the additional peaks appeared that overlapped with the europium one. Additional peaks were also noted in the presence of Al(III) and U(VI) in the solution. Despite that these peaks did not coincide with the europium signal, it was reported that the europium signal was suppressed due to the stronger adsorption of Al(III) and U(VI) complexes on the HMDE and/or their competition for cupferron. When examining the influence of other REEs (except promethium and lutetium) on the europium signal, it turned out that all lanthanides gave a peak well separated from the Eu one. However, only ytterbium gave an individual peak, while the signals from the other lanthanides overlapped. The distinct behavior of Yb can follow from the higher stability of the complex formed with cupferron, as it is well known that the stability constants of lanthanide complexes increase with increasing atomic number. Using this technique, it may be possible to detect other single REEs by combining the AdSV method with an appropriate separation method such as chromatography [24].

The paper [47] shows that among the tested ions, Mn(II), Fe(II), Mg(II), and Pb(II) exert slight interference on the Eu(III) signal received on the SDBS/LaB_6_ sensor when their concentrations are 20–50 times higher than that of Eu(III). As regards rare earth ions, Ce(III), Er(III), and La(III) have a major influence on the detection of Eu(III), even at their concentration in the solution corresponding to the concentration of europium. It is assumed that because of the great similarity in the chemical structure to Eu(III), REE ions may compete with Eu(III) for the reaction with SDBS at the LaB_6_ electrode, thus reducing the analytical signal of europium.

In the work [43], it is only mentioned that the procedure for Eu(III) determination on a gold electrode modified with 2-pyridinol-1-oxide is free from interferences from other trivalent ions (e.g., Al and Fe), which are more widespread and can also form complexes with 2-PO.

The IIPs-CPE sensor exhibits high selectivity in the determination of Eu(III), even in the presence of other lanthanide ions. Among them, Gd(III) and Sm(III) show the greatest influence on the Eu(III) signal, but a 5-fold excess of these ions does not disrupt the voltammetric response of the sensor at all. Moreover, a 10-fold excess of Cd(II) and Cu(II) significantly affects the electrode signal [41].

The paper [42] reported no serious interference from prevalent ions at their 100–500-fold excess in relation to Eu(III) concentration as well as at a 50-fold excess of other REEs (Table 4). Without a doubt, the exceptional selectivity of the Eu(III)-IIM/PC/SPE is the result of the ion-printed cavities present in Eu(III)-IIM, which are merely complementary to Eu(III).

Only in the work [44], devoted to the determination of europium at screen-printed electrodes modified with salicylamide self-assembled on mesoporous silica, no data on selectivity was found. On the other hand, in the work [43], the possibility of europium detection on the gold electrode modified with 2-pyridinol-1-oxide in the presence of other lanthanides could not be assessed, due to a lack of data.

#### 3.1.3. Selectivity of Other Procedures

The article [34] describes a new type of electrochemical sensor for the simultaneous determination of Ce(III) and Gd(III) by the double ion imprinting method. The DIIP@MNPs–SPC electrode proposed by Prasad et al. responds quantitatively to both analytes occurring simultaneously in a mixture and is not responsive to either of the interfering components when tested individually or with mixed interfering substances. The effect of potential interferents on the voltammetric response of Ce(IV) and Gd(III) is shown in Table 5.

The study [49] made an attempt to simultaneously determine La(III), Ce(III) and Pr(III) using an HMDE working electrode and o-cresolphthalexone as a complexing agent. The majority of ions do not make complexes with OCP [68], so their presence does not influence the determination of lanthanides. On the contrary, mixtures of lanthanide ions must be separated due to the similarity of the peak potentials of the complexes. The AdSV measurements carried out for the entire mixture of these ions would result in obtaining one peak current, the height of which would correspond to the content of all light lanthanides present in the sample, not to individual ones. This is similar to the work [54], concerning heavy REEs, the work [51], regarding middle and heavy rare earths, and the work [56], describing the determination of all lanthanides [54]. This was mentioned in Section 2.1.

Despite that the work [51] concerns the quantification of a series of lanthanides, the interference was examined only for holmium. It turned out that although the recorded voltammogram does not comprise a signal from the complex of coexisting ions with Alizarin, such as Ni(II), Co(II), Cu(II), Pb(II) and Zn(II), the presence of these ions in the sample reduces the peak current of the Ho(III)–ALC complex.

The selectivity of an MBTH-modified CPE towards La(III) was assessed in the presence of Al(III), Ba(III), Ce(II), and Cu(II) in the tested sample, but the La(III) signal remained unchanged [50]. Indirect determination of lutetium using the ASV method is interfered with by all ions forming EDTA complexes with comparable or greater stability than Zn-EDTA. Therefore, this method can only be used to determine Lu(III) in a sample in which these ions are not present [57].

Scandium(III) was determined at a CPE electrode using both Alizarin [52] and Alizarin S [53]. In [52], serious interference was reported in the presence of F^−^, C_2_O_4_^2−^, and citrate, while trace heavy rare earth ions, Fe(III) and Zr(IV) interfered with the Sc(III) signal in [53]. The authors propose that interference from the Fe(III) can be withdrawn using L-ascorbic acid, whereas interference caused by other metal ions can be defeated by using extraction [53].

### 3.2. Impact of Organic Compounds

The determination of rare earth elements by stripping voltammetry method is associated with interference from organic substances, which very often disrupt direct measurement and sometimes even make it impossible. This is due to the adsorption of these substances on the surface of the working electrode, which blocks active sites of the electrode and precludes the pre-concentration of the target rare earth elements. The procedures developed in [31,47,49,55] were tested for the interference of organic substances such as gelatin, albumin, cholesterol, humic acids (HA), Rhamnolipid (biosurfactant), cetyltrimethylammonium bromide (CTAB- cationic surfactant), Triton X-100 (non-ionic surfactant), as well as sodium dodecylbenzene sulfonate (SDBS) and sodium dodecyl sulfate (SDS), as the representatives for non-ionic surfactants.

When gelatin and albumin were added in an amount greater than 4 mg L^−1^, the quantitation of 1 µg L^−1^ levels of lanthanides suffered from a significant diminution of the peak current. And, conversely, the addition of a 103-fold excess of cholesterol in relation to the lanthanide concentration results in a 300 percent increase in voltammetric response. However, the quantitative usefulness of this uncommon enhancement effect is restricted due to the limited linear range. According to the authors, solutions involving a high amount (>103 times) of interfering surfactants may need a separation step prior to the analysis by the AdSV method [49]. Upon addition of 5 mg L^−1^ rhamnolipid and humic acids, respectively, 40% and 50% attenuation of the Ce(III) peak current was observed, while at the same concentration of CTAB, the signal of Ce(III) disappeared completely. To address this issue, premixing the sample with Amberlite XAD-7 resin before the AdSV measurement was recommended. It was found that due to premixing with Amberlite XAD-7 resin, CTAB, rhamnolipid, and HA did not interfere with the peak current in a wide range of concentrations (from 0.5 to 10 mg L^−1^) [31].

Upon addition of Triton X-100 at a concentration equal to 0.7 µg mL^−1^, the peak of the Yb(III)-TTA-PEG complex was depressed, whereas, in the presence of 3 µg mL^−1^ of Triton X-100, the peak current of Yb(III) totally decayed. In order to avoid interference from surfactants, the authors suggest exposing water samples tested to ultraviolet irradiation, optionally with the addition of hydrogen peroxide or mineralizing them [55].

In the work [47], CTAB leads to a decrease in the Eu(III) signal, whereas in the presence of SDBS and SDS, peak enhancement is observed. Concomitantly, Triton X-100 and CPB do not show any serious effect on the voltammetric response of Eu(III) (Figure 1). 

## 4. Practical Application

In most of the papers reviewed, the developed procedures for voltammetric determination of rare earth elements were checked for their analytical applicability [29,30,31,32,33,34,35,36,37,38,41,42,45,47,48,50,51,52,53]. The data obtained during the practical application of these procedures are summarized in Table 3, Table 4 and Table 5. Rare earth element ions were studied in samples of various origins, such as food, environmental, and biological samples as well as in certified reference materials. Based on the results collected in the table, the analytical usefulness of the developed procedures can be assessed.

Table 6 presents data on the use of the reviewed voltammetric procedures for quantification of REE contents in certified reference materials. The procedures [29,30,32,33] were used for analysis of the cerium content in standard samples of rare earth nodular graphite cast iron from the Chinese Ministry of Metallurgical Industry. It is worthwhile to mention that due to possible interference from Fe(III), the nodular graphite cast iron underwent a solvent extraction separation process using 1-phenyl-3-methyl-4-benzoyl-5-benzopyrazolone (PMBP) in benzene prior to the analysis. It can be seen that in each case the results obtained are in good agreement with the certified values, which proves the accuracy of all developed procedures and confirms the possibility of using them for analysis of nodular cast iron. Rare earth nodular graphite cast iron standard samples, such as BH1902-1, BH1905-1, and JC 79-18, were also used in the work [51] to examine recoveries of the sum of REEs, except La, Ce, Pr, Nd, and Sc. The recovery values ranging from 92.9 to 97.8% confirmed that the method based on the use of Alizarin and a carbon paste electrode is analytically useful [51]. On the other hand, NIES CRM no. 8 vehicle exhaust particulates were used to check the analytical suitability of the europium determination procedure developed using an N/MWCNTs/GCE. The average detectable Eu(III) concentration was less than the reference value due to other rare earth elements co-existing in the samples, but the recovery values between 91.6 and 108.9% for Eu(III) clearly testify to the applicability of this procedure for detection of trace Eu(III) in vehicle exhaust particulate samples [45].

Table 7 shows the results obtained during the determination of REEs by voltammetry methods in environmental and biological samples. As natural samples, real water samples, such as drinking water, seawater, river water, lake water, and tap water, as well as red tribasic fluorescent powder samples and mineral samples, were selected. Synthetic water samples were also investigated. The biological samples tested included human urine and human blood serum. Among the tested samples, only human urine and the red tribasic fluorescent powder sample contained the REEs determined. The remaining samples were enriched with the determined lanthanides prior to the analysis and were used to examine the recoveries of these elements to check the accuracy of the developed sensors. Mineral samples were submitted to an extraction process by PMBP prior to the analysis [52,53]. As can be seen, the obtained results and recovery values demonstrate that all the tested voltammetric methods are reliable and accurate. Additionally, the results of the voltammetric determination of Ce(III) and Eu(III), both in natural and synthetic water samples, were compared with those obtained by the ICP method and they were found to be in agreement [33,41]. In the work [48], due to the poor peak resolution on the GCE/SbF sensor, instead of simultaneous analysis of Ce(III), La(III), and Pr(III) in tap water, a single REE analysis was performed. In this procedure, the best recovery values were obtained for cerium (96.60–97.28%), while the lowest ones for praseodymium (81.48–84.80%), which is due to the fact that the developed sensor shows the greatest sensitivity to cerium and the lowest one to praseodymium. Special attention should be paid to very low standard deviation values <1.3 and recoveries close to 100% obtained during simultaneous quantification of Ce(III) and Gd(III) in real water samples and human blood serum [34]. These validation data, indicating the excellent accuracy of the sensor developed in this procedure, namely a screen-printed carbon electrode modified with double-ion imprinted polymer @magnetic nanoparticles, are undoubtedly associated with the presence of double-ion imprinted polymer nanoparticles in the carbon paste electrode composition. The high affinity of the imprinted polymers both to cerium and gadolinium ions has an undeniable impact on the observed advantages of this sensor. Therefore, this sensor can be successfully used as a dependable tool for the simultaneous determination of Ce(III) and Gd(III) in real water samples as well as in biological samples such as human blood serum, without tedious separation [34]. An ion-imprinted polymer-based sensor was also developed in the work [35] and its analytical usefulness was verified during seawater analysis. It is worth mentioning that seawater, due to the complex matrix rich in salt, causes both spectral disturbances and matrix effects for powerful techniques such as ICP-MS. Therefore, is hard to perform the quantification of rare earths at ultratrace levels in seawater by these methods without a previous separation of the matrix components and preconcentration of the analytes. Nevertheless, Alizadeh et al. managed to develop a highly selective nano-Ce-imprinted polymer-modified voltammetric sensor, the use of which allowed for the interference-free determination of the Ce(III) level in seawater, omitting sample pre-treatment with satisfied results. 

Table 8 reports the data obtained during the analysis of food and environmental samples by stripping voltammetry and those obtained by the ICP-OES method. All these samples showed a natural content of the tested lanthanides. Therefore, there was no need to enrich them in the tested lanthanides, unlike the analyzed samples included in Table 3. Moreover, all these samples were analyzed using two independent methods, such as AdSV and ICP-OES. In food, phosphate, and catalyst samples as well as waste water and Ce(III)-polluted water samples the content of cerium was measured, whereas sand samples were used to determine lanthanum. The paper [38] reports that the content of cerium was determined in food samples such as spinach, mushroom, rice, and Pu’er tea to evaluate the performance of a poly(catechol) (PC) film glassy carbon electrode (GCE) modified with Ce(III) ion-imprinted membrane (IIM). The highest cerium content was found in mushrooms, while the lowest one was in Pu’er tea. The content of lanthanum was measured using an MBTH-modified CPE in different monazite sand samples obtained from Bangka Belitung [50]. In the work [33], cerium was detected in two different samples of phosphate concentrate from the Esfodi Phosphate Industrial & Mineral Complex in Yazd, Iran. On the other hand, catalyst samples were used to rate the applicability of the cerium determination method developed using IIPs-CPE [36]. A comparison with the ICP-OES method demonstrated that all developed voltammetric methods give reliable results and thus they are suitable for the analysis of real samples with a complicated matrix. 

## 5. Conclusions

Rare earth elements are hard to determine using electrochemical methods because the reduction of cations of these elements to the metallic state requires the use of great negative potentials, at which the supporting electrolyte decomposes. Therefore, when using stripping voltammetry methods, effective preconcentration can mainly be achieved by non-electrolytic preconcentration techniques. Hence, as the conclusions of this work show, the researchers practically used only the adsorption stripping voltammetry method for the determination of REEs, with two exceptions. One exception concerns the indirect ASV determination of lutetium(III) based on the electroanalytical reaction of zinc, whilst the second one applies the quantification of cerium(III) by CSV in which insoluble CeO_2_ accumulated on an indium tin oxide electrode is reduced to Ce(III). Adsorption stripping voltammetric determination of lanthanides was possible owing to the use of a complexing agent added directly to the tested sample, electrochemically accumulated on the electrode surface in the form of a film/membrane or incorporated into the electrode composition. Undeniably, Alizarin was the most commonly used complexing agent added directly to the tested samples, whereas poly(catechol) was the most frequently used ligand in the form of a film on the surface of the working electrode. Another significant factor in voltammetric procedures is the type of working electrode used on which the electroanalytical reaction being the basis of the measurement is performed. Fortunately, only a few voltammetric procedures for lanthanide quantification are based on the use of mercury electrodes. In the vast majority of the works, the working electrodes used are mercury-free solid carbon electrodes, which fits perfectly into the assumptions of Green Analytical Chemistry. Most procedures have been developed using a carbon paste or glassy carbon electrode. Many authors made numerous modifications to these electrodes, which usually significantly decreased the detection limits and improved the selectivity. As shown in Table 1, the lowest detection limits equal to 1 × 10^−12^ M was achieved in the methods based on the use of modified electrodes [38,50]. It is worth mentioning that in practically all procedures dedicated to REE determination, the effect of many foreign ions as potential interfering factors of the voltammetric response of lanthanides was checked. Most of the reviewed procedures were tested for interference from foreign ions commonly present in real samples and it was also checked whether it was possible to determine individual lanthanides in a mixture of other lanthanides. As data in Table 1 show, in most methods of voltammetric determination of several REEs concomitantly, it is not possible for all these lanthanides to be present in the sample when determining one of them, because the peak potentials of these elements are similar and the peaks would overlap. This applies to methods [48,49,51,54,56]. Therefore, it is necessary to separate the lanthanide ion mixtures before the voltammetric analysis is performed using these procedures. On the other hand, methods [34,46] can be used for simultaneous determination of Ce(IV) and Gd(III), as well as Eu(III) and Yb(III), respectively, because differences in their peak potentials allow for obtaining satisfactory resolution of both peaks. The susceptibility of the REE signal to disruptions from the tested interferents depends largely on both the working electrode and the complexing agent used in the developed procedures. As for the effect of foreign ions, other than lanthanides, it can be noted that they usually occur in those procedures in which the complexing agent is added directly to the tested sample, but they are not significant. The type of chemical modifier of the electrode used also has a significant impact on selectivity. Carbon paste electrodes modified with ion-imprinted polymers deserve special attention because they are distinguished by high selectivity due to specific sites in the polymer that can ensure strong coordination only with specific lanthanides. By using these modified electrodes, the detection of individual lanthanides in a mixture is possible. This is a great achievement, considering the similarity of the electrochemical properties of lanthanides and the difficulties in separating REEs using electroanalytical techniques. Otherwise, the authors of only four articles examined the impact of organic substances constituting the matrix of real samples on the possibility of determining rare earth elements by the developed voltammetry method. The influence of organic substances commonly present in biological samples, such as gelatin, albumin, and cholesterol as well as humic acids (HA) and surfactants (CTAB, Triton X-100, SDBS, SDS) constituting matrices of environmental samples, was studied. The above substances influence the voltammetric response of lanthanides, usually leading to its reduction or complete disappearance. The practical applications of most of the developed voltammetric procedures are collected in Table 6, Table 7 and Table 8. The reviewed methods were checked for their analytical usefulness by applying them to analyze lanthanides in food, environmental, and biological samples, as well as in certified reference materials. The data collected in the tables confirm that each of the developed procedures is correct and appropriate for the determination of lanthanides in real samples. To sum up, it can be stated that the present review of the voltammetric procedures dedicated to the quantification of rare earth elements is a concise and comprehensive summary of their analytical characteristics and applicability.

## Figures and Tables

**Figure 1 molecules-28-07755-f001:**
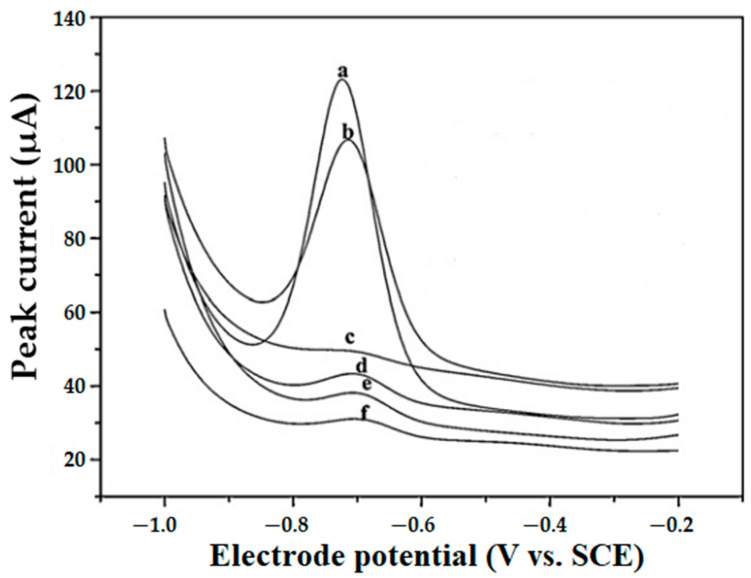
Effect of various surfactants on the DP voltammograms of 10 µM Eu^3+^: (a) SDBS (1.2 mM), (b) SDS (8.0 mM), (c) CTAB (0.1 mM), and (d) Trion X-100 (3.1 mM), (e) no surfactant, and (f) CPB (0.05 mM). After re-elaboration from Ref. [47] and permission of the publisher.

**Table 1 molecules-28-07755-t001:** Analytical performance of voltammetric procedures for rare earth elements determination. The methods were ranked by means of rising limit of detection.

Tested Ion	Method	Working Electrode	Complexing Agent	*LOD* (M)	Accumulation Time (s)	Peak Potential *E*p (V)	Linear Range (M)	Investigated Interferents			Ref.
								Foreign Ions (Other Than REEs)/Organics: Interfering	Foreign Ions (Other Than REEs)/Organics: No Interfering	REEs: Interfering/No Interfering	
La(III)	AdSV	MBTH/CPE	-	1.0 × 10^−12^	-	−0.22 (vs. Ag/AgCl, 3 M KCl)	1.0 × 10^−12^– 7.0 × 10^−11^	-	Al(III), Ba(II), Cu(II)	-/ Ce(III)	[50]
Ce(III)	AdSV	Ce-IIM/PC/GCE	-	1.0 × 10^−12^	600	0.88 (vs. Ag/AgCl, 3 M KCl)	3.0 × 10^−12^– 1.0 × 10^−4^	Cu(II), Fe(III), Ni(II)	Co(II), Mg(II), Na(I), Zn(II)	-/ Dy(III), Er(III), Eu(III), Gd(III), Ho(III), Nd(III), Pr(III), Tb(III), Yb(III)	[38]
Ce(III)	AdSV	Ce-IP/MWCNT/CPE	-	1.0 × 10^−11^	-	1.05 (vs. Ag/AgCl, 3 M KCl)	2.5 × 10^−11^– 1.0 × 10^−6^	-	Ag(I), Cr(III), Cd(II), Co(II), Hg(II)	Dy(III), Eu(III) / La(III), Nd(III), Sm(III), Tb(III), Yb(III)	[35]
Eu(III)	AdSV	HMDE	Cupferron	6.0 × 10^−11^	60	−0.88 (vs. Ag/AgCl, 3 M KCl)	0–1.3 × 10^−8^	Cr(III)	Al(III), Mo(VI), U(VI), V(V)	-/ Dy(III), Er(III), Eu(III), Gd(III), Ho(III), Nd(III), Pr(III), Sm(III), Tb(III), Yb(III)	[40]
Dy(III) Ho(III) Er(III) Tm(III) Yb(III) Lu(III)	AdSV	CPE	Alizarin	1.0 × 10^−10^	60	0.586 0.588 0.588 0.584 0.582 0.580 (vs. SCE)	1.0 × 10^−9^– 2.0 × 10^−7^	Co(II), Cu(II), Ni(II), Pb(II), Zn(II)	Ca(II), Ba(II), Cr(III), Se(IV), B(III), Ge(IV), As(III), Ag(I), Mn(II),Mg(II),Cd(II), Al(III), V(V), Hg(II), Ti(IV), Sb(III), Sn(IV), Fe(II), Ga(III), Fe(III), Th(IV), Zr(IV), In(III), SO_4_^2−^, PO_4_^3−^, F^−^	-/ La(III), Ce(III), Pr(III), Nd(III), Sc(III)	[51]
La(III) Ce(III) Pr(III)	AdSV	SMDE	OCP	1.2 × 10^−10^ 1.7 × 10^−10^ 1.4 × 10^−10^	1200	−0.95 −1.00 −1.05 (vs. Ag/AgCl, 3 M KCl)	2.5 × 10^−9^– 2.5 × 10^−8^	gelatin, albumin	Ca(II), Mg(II), Al(III), Cu(II), Cd(II), Hg(II), Zn(II), cholesterol, chloride	no data	[49]
Ce(III)	AdSV	PC/GCE	-	2.0 × 10^−10^	10	0.85 (vs. Ag/AgCl, 3 M KCl)	2.0 × 10^−9^– 1.0 × 10^−7^	Al(III), Bi(III)	Zn(II), Cu(II), Pb(II), Cd(II), Hg(II), Tl(I), Re(II), Sb(III), Ge(IV),Te(IV), Se(IV), Ag(I), Au(I), Sn(IV), Co(II)	no data	[37]
La(III) Ce(III) Pr(III)	AdSV	GC/SbFE	Alizarin	3.0 × 10^−9^ 4.3 × 10^−10^ 5.0 × 10^−9^	360	0.74 0.76 0.79 (vs. Ag/AgCl, 3 M KCl)	7.1 × 10^−9^– 1.8 × 10^−7^	-	Co(II), Fe(II), Mn(II), Ni(II), Pb(II), Zn(II)	La(III)/-	[48]
Ce(IV) Gd(III)	AdSV	DIIP@MNPs/SPCE	-	5.0 × 10^−10^ 1.2 × 10^−9^	180	0.05 −0.37 (vs. Ag/AgCl, 3 M KCl)	1.9 × 10^−9^– 3.8 × 10^−8^ 4.6 × 10^−9^– 5.5 × 10^−8^	-	Cr(III), As(III), Ca(II), Mg(II), Al(III), Fe(III), SO_4_^2−^, PO_4_^3−^, ascorbic acid	-/ Dy(III), Ho(III), Nd(III), Pr(III), Y(III)	[34]
La(III) Tb(III) Yb(III)	AdSV	SMDE	MR19	8.0 × 10^−10^ 5.0 × 10^−10^ 5.0 × 10^−10^	60	−0.682 −0.754 −0.784 (vs. Ag/AgCl, 3 M KCl)	1.0 × 10^−8^– 1.0 × 10^−6^	no data			[54]
Y(III) Dy(III) Ho(III) Yb(III)	AdSV	HMDE	SVRS	1.4 × 10^−9^ 1.1 × 10^−9^ 1.0 × 10^−9^ 5.0 × 10^−10^	180	−0.98 −0.98 −1.00 −1.00 (vs. Ag/AgCl, 3 M KCl)	0–3.4 × 10^−7^ 0–2.5.× 10^−7^ 0–1.8 × 10^−7^ 0–2.3 × 10^−7^	no data			[56]
Ce(III)	AdSV	CTAB/CPE	Alizarin	6.0 × 10^−10^	120	0.73 (vs. SCE)	8.0 × 10^−10^– 8.0 × 10^−9^	-	Ca(II), Ba(II), B(III), As(III), Mg(II), Se(IV), Ge(IV), Mn(II), Zn(II), Cr(III), Ni(II), Hg(II), Cd(II), Co(II), Fe(II), Pb(II), Cu(II), Al(III), Bi(III), Fe(III), Zr(IV), In(III), Ga(III), HCr_2_O_7_^−^, MnO_4_^−^, AuCl_4_^−^, SO_4_^2−^, PO_4_^3−^, F^−^, ascorbic acid	-/ La(III), Pr(III), Nd(III), Sm(III), Eu(III), Y(III), Gd(III), Tb(III), Sc(III), Dy(III), Ho(III), Er(III), Yb(III), Tm(III)	[30]
Sc(III)	AdSV	CPE	Alizarin	6.0 × 10^−10^	60	−0.60 (vs. SCE)	1.0 × 10^−9^– 6.0 × 10^−7^	F^−^, C_2_O_4_^2−^, citrate	Ca(II), Mg(II), Zn(II), Cd(II), Mn(II), Ag(I), As(III), Au(III), Ba(II), Co(II), Cr(III), Hg(II), Ni(II); MoO_4_^2−^, Pb(II), Al(III), Sn(II), Ga(III) Cu(II), Fe(III), Sb(III), V(V), In(III); Bi(III), Th(IV), Zr(IV), Ti(IV), SO_4_^2−^, PO_4_^3−^	-/ Ce(III), Dy(III), Er(III), Eu(III), Gd(III), Ho(III), La(III), Nd(III), Pr(III), Tb(III), Yb(III)	[52]
Sc(III)	AdSV	CPE	Alizarin S	6.0 × 10^−10^	180	−0.58 (vs. SCE)	1.0 × 10^−9^– 4.0 × 10^−7^	Fe(III), Zr(IV)	Zn(II), Pb(II), Ni(II), Li(I), Co(II), Mn(II), Cr(III), As(III), Se(IV), Ag(I), Au(III), Be(II), Bi(III), Cd(II), Ga(III), Fe(II), Mo(VI), Sn(II), Cu(II), Ba(II), V(V), CNS^−^, SO_4_^2−^, PO_4_^3−^, F^−^, CN^−^	Eu(III), Gd(III), Tb(III), Dy(III), Ho(III), Er(III), Tm(III), Yb(III), Lu(III) / La(III), Ce(III), Pr(III), Nd(III), Sm(III)	[53]
Ce(III)	AdSV	NHMF/CPE	-	8.0 × 10^−10^	350	0.55 (vs. Ag/AgCl, 3 M KCl)	5.0 × 10^−9^– 9.0 × 10^−8^	-	Cd(II), Cr(III), Cu(II), Mn(II), Ni(II), Pb(II), Th(IV), Zn(II), Br^−^, Cl^−^, SO_4_^2−^, CH_3_COO^−^, UO_2_^2+^	La(III), Sm(III) / Er(III), Ho(III)	[33]
Ce(III)	AdSV	CPE	Alizarin	2.0 × 10^−9^	120	0.69 (vs. SCE)	6.0 × 10^−9^– 3.0 × 10^−7^	Th(IV)	Ba(II), Ca(II), Cr(III), Mg(II), As(III), Se(IV), B(III), Ge(IV), Mn(II), Cd(II), Pb(II), In(III) Co(II), Zn(II), V(V), Hg(II), Fe(III), Fe(II), Ni(II), Sn(IV), Sb(III), Ti(IV), Al(III), Zr(IV), Cu(II), Bi(III) Ga(III), SO_4_^2−^, PO_4_^3−^, ascorbic acid	-/ La(III), Pr(III), Nd(III), Sm(III), Eu(III), Y(III), Gd(III), Tb(III), Sc(III), Dy(III), Ho(III), Er(III), Yb(III), Tm(III)	[29]
Lu(III)	ASV	HMDE	-	2.1 × 10^−9^	120	−0.995 (vs. Ag/AgCl, 3 M KCl)	2.1 × 10^−9^– 7.3 × 10^−6^	no data			[57]
Ce(III)	AdSV	DPNSG/CPE	-	2.3 × 10^−9^	600	0.27 (vs. Ag/AgCl, 3 M KCl)	2.30 × 10^−9^– 6.45 × 10^−8^	-	Cd(II), Cr(III), Cu(II), Mn(II), Ni(II), Pb(II), Th(IV), Zn(II), Br^−^, Cl^−^, CH_3_COO^−^, SO_4_^2−^, UO_2_^2+^,	La(III) / Er(III), Ho(III), Sm(III)	[32]
Ce(III)	CSV	ITO electrode	-	5.8 × 10^−9^	300	0.55 (vs. Ag/AgCl, 3 M KCl)	1.0 × 10^−7^– 7.0 × 10^−7^	Mn(II)	Bi(III), Cu(II), Zn(II), Sn(II), Mg(II)	-/ Eu(III)	[39]
Eu(III)	AdSV	SDBS/LaB_6_ electrode	-	6.0 × 10^−9^	120	−0.70 (vs. SCE)	1.0 × 10^−8^– 2.0 × 10^−6^	Fe(II), Mg(II), Mn(II), Pb(II), SDBS, SDS, CTAB	Na(I), Ca(II), Zn(II), Triton X-100, CPB	Ce(III), Er(III), La(III) / Sm(III), Yb(III)	[47]
Yb(III)	AdSV	HMDE	TTA-PAG ligand	-	180	−1.65 (vs. Ag/AgCl, 3 M KCl)	5.0 × 10^−9^– 1.0 × 10^−7^	-	Cr(III), Co(II), Mn(II), Mo(VI), U(VI), V(V), Triton X-100	-/ Eu(III), La(III), Y(III)	[55]
Eu(III)	AdSV	N/MWCNTs/GCE	-	1.0 × 10^−8^	60	−0.70 (vs. SCE)	4.0 × 10^−8^– 1.0 × 10^−4^	Bi(III), Cr(III)	Mn(II), Co(II), Pd(II), Mg(II), Zn(II), Fe(II), Ba(II), Ni(II)	Er(III), La(III), Sm(III), Yb(III) /-	[45]
Eu(III)	AdSV	Sal-SAMMS/SPCE	-	1.0 × 10^−8^	300	−0.75 (vs. Ag/AgCl, 3 M KCl)	7.5 × 10^−8^– 5.0 × 10^−7^	no data			[44]
Eu(III) Yb(III)	AdSV	NCTMFE	-	3.0 × 10^−8^ 2.0 × 10^−8^	300	−0.62 −1.46 (vs. SCE)	X– 2.0 × 10^−6^	no data		La(III)/no data	[46]
Ce(III)	AdSV	GCE	Alizarin S	6.0 × 10^−8^	30	0.60 (vs. Ag/AgCl, 3 M KCl)	2.0 × 10^−7^– 8.0 × 10^−6^	Cr(III), Fe(II), Sb(III), V(V)	Al(III), As(III), As(V), Cd(II), Co(II), Cr(VI), Hg(II), K(I), Mg(II), Mn(II), Na(I), Ni(II), Pt(IV), Se(IV), Se(VI), Sn(II), Ti(IV), U(VI), Zn(II), Bi(III), Ga(III), Cu(II), Mo(VI), CTAB, rhamnolipid, humic acid, Triton X-100, SDS, fulvic acid, natural organic matter	no data	[31]
Eu(III)	AdSV	IIM/PC/SPE	-	1.0 × 10^−7^	300	−1.00 (vs. Ag/AgCl, 3 M KCl)	3.0 × 10^−7^– 1.0 × 10^−3^	-	Ca(II), Co(II), Cu(II), Fe(III), Mg(II), Na(I), Ni(II), Zn(II)	-/ Dy(III), Er(III), Ce(III), Gd(III), Ho(III), Nd(III), Pr(III), Tb(III), Yb(III)	[42]
Eu(III)	AdSV	IIPs-CPE	-	1.5 × 10^−7^	20	−0.18 (vs. Ag/AgCl, 3 M KCl)	5.0 × 10^−7^– 3.0 × 10^−5^	Cd(II), Cu(II)	Ag(I), Ca(II), Hg(II), Mg(II), Pt(II), Zn(II)	Ce(III), Gd(III), Sm(III) / Er(III), Dy(III), La(III)	[41]
Ce(III)	AdSV	IIPs-CPE	-	1.5 × 10^−7^	20	0.93 (vs. Ag/AgCl, 3 M KCl)	1.0 × 10^−6^– 1.0 × 10^−5^	no data		-/ Dy(III), Er(III), Eu(III), Gd(III), Ho(III), Nd(III), Pr(III), Tb(III), Yb(III)	[36]
Eu(III)	AdSV	PO/GE	-	3.0 × 10^−7^	-	1.10 (vs. Ag/AgCl, 3 M KCl)	1.0 × 10^−6^– 8.0 × 10^−5^	-	Al(III), Fe(III)	no data	[43]

AdSV—Adsorptive stripping voltammetry, CPE—Carbon Paste Electrode, MBTH—3-Methyl-2-hydrazinobenzothiazole, GCE—Glassy carbon electrode, Ce-IIM—cerium ion-imprinted membrane, PC—poly-catechol, Ce-IP—cerium-imprinted polymer, MWCNT—multiwall carbon nanotubes, HMDE—Hanging Mercury Drop Electrode, SMDE—Static Mercury Drop Electrode, OCP—o-cresolphthalexon, GC/SbFE—Glassy carbon antimony film electrode, ASV—Anodic stripping voltammetry, DIIP@MNPs/SPCE—Screen-printed carbon electrode modified with double ion-imprinted polymer @ magnetic nanoparticles, MR19—Mordant Red 19, SVRS—Solochrome Violet RS, CTAB—Cetyltrimethylammonium bromide, NHMF—N′-[(2-hydroxyphenyl)methylidene]-2-furohydrazide, DPNSG—dipyridyl-functionalized nanoporous silica gel, CSV—Cathodic stripping voltammetry, ITO—Indium tin oxide, SDBS/LaB_6_—sensor based on lanthanum hexaboride and sodium dodecylbenzene sulfonate, TTA-PAG ligand—mixed complex of 2-thenoyltrifluoroacetone and polyethyleneglycol, N/MWCNTs—multiwall carbon nanotubes and Nafion composite film, Sal-SAMMS/SPCE—Screen-printed carbon electrode modified with salicylamide self-assembled monolayers on mesoporous silica, NCTMFE—Nafion-coated thin mercury film electrode, IIM/PC/SPE—Screen-printed electrode modified with ion-imprinted membrane and poly-catechol, IIPs—ion-imprinted polymers, PO/GE—gold electrode modified with 2-pyridinol-1-oxide, SCE—saturated calomel electrode.

**Table 2 molecules-28-07755-t002:** Effect of potential interferents on the voltammetric response of Ce(III) measured on Ce(III)-IIM/PC/GCE. All measurements were carried out in a solution containing 1 × 10^−6^ M Ce(III). After re-elaboration from Ref. [38] and permission of the publisher.

Interferent	Interferent Concentration (M)	Peak Current of 1 × 10^−6^ M Ce(III) (µA)
-	-	2.60
Na^+^	5.0 × 10^−4^	2.60
Ca^2+^	5.0 × 10^−4^	2.60
Mg^2+^	5.0 × 10^−4^	2.55
Zn^2+^	5.0 × 10^−5^	2.48
Ni^2+^	5.0 × 10^−5^	2.35
Fe^3+^	5.0 × 10^−5^	2.10
Co^2+^	5.0 × 10^−5^	2.50
Cu^2+^	5.0 × 10^−5^	2.35
Ho^3+^	2.0 × 10^−5^	2.60
Dy^3+^	2.0 × 10^−5^	2.60
Er^3+^	2.0 × 10^−5^	2.50
Eu^3+^	2.0 × 10^−5^	2.60
Gd^3+^	2.0 × 10^−5^	2.55
Pr^3+^	2.0 × 10^−5^	2.60
Nd^3+^	2.0 × 10^−5^	2.55
Tb^3+^	2.0 × 10^−5^	2.60
Yb^3+^	2.0 × 10^−5^	2.60

**Table 3 molecules-28-07755-t003:** Effect of potential interferents on the voltammetric response of Ce(III) measured on Ce-IIPs-CP. All measurements were carried out in a solution containing 1 × 10^−5^ M Ce(III). After re-elaboration from Ref. [36] and permission of the publisher.

Interferent	Interferent Concentration (M)	Peak Current of 1 × 10^−5^ M Ce(III) (µA)
-	-	0.880
Ho^3+^	5.0 × 10^−3^	0.810
Dy^3+^	5.0 × 10^−3^	0.790
Er^3+^	5.0 × 10^−3^	0.822
Eu^3+^	5.0 × 10^−3^	0.820
Gd^3+^	5.0 × 10^−3^	0.821
Pr^3+^	5.0 × 10^−3^	0.821
Nd^3+^	5.0 × 10^−3^	0.817
Tb^3+^	5.0 × 10^−3^	0.808
Yb^3+^	5.0 × 10^−3^	0.820

**Table 4 molecules-28-07755-t004:** Effect of potential interferents on the voltammetric response of Eu(III) measured on Eu(III)-IIM/PC/SPE. All measurements were carried out in a solution containing 1 × 10^−5^ M Eu(III). After re-elaboration from Ref. [42] and permission of the publisher.

Interferent	Interferent Concentration (M)	Peak Current of 1 × 10^−5^ M Eu(III) (µA)
-	-	1.50
Na^+^	5.0 × 10^−3^	1.50
Ca^2+^	5.0 × 10^−3^	1.50
Mg^2+^	5.0 × 10^−3^	1.50
Zn^2+^	1.0 × 10^−3^	1.50
Ni^2+^	1.0 × 10^−3^	1.50
Fe^3+^	1.0 × 10^−3^	1.50
Co^2+^	1.0 × 10^−3^	1.49
Cu^2+^	1.0 × 10^−3^	1.48
Ho^3+^	5.0 × 10^−4^	1.50
Dy^3+^	5.0 × 10^−4^	1.48
Er^3+^	5.0 × 10^−4^	1.50
Eu^3+^	5.0 × 10^−4^	1.50
Gd^3+^	5.0 × 10^−4^	1.49
Pr^3+^	5.0 × 10^−4^	1.50
Nd^3+^	5.0 × 10^−4^	1.49
Tb^3+^	5.0 × 10^−4^	1.50
Yb^3+^	5.0 × 10^−4^	1.50

**Table 5 molecules-28-07755-t005:** Effect of potential interferents on the voltammetric response of Ce(IV) and Gd(III) measured on DIIP@MNPs-SPCE. All measurements were carried out in a solution containing 2 ng mL^−1^ both Ce(IV) and Gd(III) as well as 200 ng mL^−1^ each interferents. After re-elaboration from Ref. [34] and permission of the publisher.

Interferent	Ce(IV)			Gd(III)		
	Peak Current (µA)	Recovery (%)	RSD (%) (*n* = 3)	Peak Current (µA)	Recovery (%)	RSD (%) (*n* = 3)
Ho^3+^	1.078	100	0.28	6.80	99.6	2.5
Nd^3+^	1.000	92.8	2.00	6.40	93.7	4.02
Y^3+^	1.082	100.4	0.34	6.78	99.3	0.65
Dy^3+^	0.959	89.0	1.20	6.52	95.5	1.36
Pr^3+^	1.020	94.6	1.06	6.62	96.9	0.68
Cr^3+^	1.050	97.4	0.39	6.43	94.1	0.79
As^3+^	1.040	96.5	0.58	6.82	99.8	1.44
Ca^2+^	1.060	98.3	1.23	6.79	99.4	2.36
Mg^2+^	1.070	99.3	2.25	6.69	97.9	0.39
Al^3+^	1.098	101.8	1.10	6.80	99.6	2.58
Fe^3+^	0.998	98.3	0.35	6.43	94.1	3.58
Ascorbic acid	1.072	99.4	2.63	6.78	99.4	0.84
PO_4_^3−^	1.068	99.1	1.56	6.80	99.7	1.56
SO_4_^2−^	1.088	100.9	0.71	6.72	98.5	2.94
Mixture of interferents	1.050	97.4	0.39	6.40	93.7	4.02

**Table 6 molecules-28-07755-t006:** Determination of REE contents in certified reference materials by voltammetry method.

Element	Sample	Certified Reference (%)	Found (%)	Ref.
Ce	BH1902-1	0.0102	0.0098 ± 0.0004	[29]
	BH1905-1	0.0115	0.0118 ± 0.0003	
Ce	BH1902-1 BH1905-1	0.0102 0.0115	0.0107 ± 0.0005 0.0112 ± 0.0004	[30]
Ce	nodular cast iron 1	0.0151	0.0146 ± 0.0015	[32]
	nodular cast iron 2	0.0109	0.0118 ± 0.0012	
Ce	nodular cast iron 1	0.0151	0.0148 ± 0.0008	[33]
	nodular cast iron 2	0.0109	0.0112 ± 0.0006	
**Element**	**Sample**	**Recovery (%)**	**Found (%)**	**Ref.**
Sm, Eu, Gd, Tb, Dy, Ho, Er, Tm, Yb, Lu	BH1902-1	97.8	0.082 ± 0.004	[51]
	BH1905-1	95.4	0.053 ± 0.002	
	JC 79-18	92.9	0.039 ± 0.002	
**Element**	**Sample**	**Certified reference (nM)**	**Found (nM)**	**Ref.**
Eu	NIES CRM no. 8 vehicle exhaust particulates A	106.0 ± 5.1	102 ± 5.0	[45]
	NIES CRM no. 8 vehicle exhaust particulates B	126.0 ± 5.8	123 ± 3.9	
	NIES CRM no. 8 vehicle exhaust particulates C	134.0 ± 5.2	131 ± 4.4	
	NIES CRM no. 8 vehicle exhaust particulates D	143.0 ± 6.8	146 ± 6.5	

**Table 7 molecules-28-07755-t007:** Determination of REEs in different environmental and biological samples by voltammetry method. Evaluation of accuracy by adding analytes to the samples and evaluating recovery.

Element	Unit	Sample	Added	Found		RSD (%)	Recovery (%)	Ref.
Sc	µg	Mineral sample 1 Mineral sample 2 Mineral sample 3	0.0080 0.0060 0.0040	0.0076 0.0055 0.0042		1.32 3.56 4.78	95.0 91.7 105	[52]
Sc	µg	Mineral sample 1 Mineral sample 2 Mineral sample 3	0.0080 0.0060 0.0050	0.0072 0.0058 0.0052		0.89 4.07 5.00	90.0 96.7 104	[53]
Ce	nM	Drink water Sea water	100.0 1.0 100.0 1.0	97.3 0.95 102.7 1.04		3.5 3.3 3.8 4.2	97.3 95.0 102.7 104.0	[35]
Ce	nM	Bystrzyca river water Lake Zemborzyce	300.0 600.0 300.0 600.0	294.6 614.4 292.2 604.8		3.7 4.2 4.0 4.9	98.2 102.4 97.4 100.8	[31]
Ce	nM	Human urine	0 2.0	2.14 4.3		5.12 1.91	- 108	[37]
Ce La Pr Ce La Pr	nM	Tap water 01 Tap water 02	71.4 72 71 357 360 355	69.0 66.0 60.2 347.0 338.0 289.1		2.75 3.33 7.00 1.10 2.66 6.70	96.60 91.67 84.79 97.20 93.88 81.44	[48]
Eu	nM	Red tribasic fluorescent powder sample	0 150	210 350		3.4 4.0	- 93	[47]
Eu	µM	Tap water Greenlake water Panlong river water	0 2.0 50 150 0 2 .0 50 150 0 2.0 50 150	0 2.0 50 150 0 2.0 50 150 0 2.0 50 150		- 2.6 2.3 1.8 - 3.0 1.2 2.5 - 3.6 2.8 1.5	- 98.5 97.6 101.5 - 99.5 98.6 100.6 - 102.5 101.6 99.7	[42]
**Element**	**Unit**	**Sample**	**Linear range**	**Recovery (%)**		** *LOD* **	**RSD** **(%)**	
Ce Gd Ce Gd Ce Gd	ng mL^−1^	Aqueous Waste water Human blood serum	0.25–6.23 0.74–9.47 0.25–6.23 0.78–8.96 0.25–5.72 0.76–9.23	98.2–103.3 97.4–103.2 98.2–103.3 97.5–102.9 98.8–102.5 98.4–101.7		0.073 0.196 0.075 0.173 0.068 0.178	0.51 0.76 1.23 0.98 0.82 1.02	[34]
**Element**	**Unit**	**Sample**	**Added**	**Found**		**RSD** **(%)**	**Recovery (%)**	**Ref.**
				**AdSV**	**ICP**			
Eu	nM	Synthetic water (Na^+^, Ca^2+^, Mg^2+^, SO_4_^2−^, Cl^−^) Tap water River water	2000 5000 1000 700	2070 4700 900 820	1880 5210 1100 780	3.7 3.5 4.2 4.6	103.5 94.0 90.0 108.7	[41]

**Table 8 molecules-28-07755-t008:** Accuracy evaluation of the voltammetric method for determination of REEs of some food and environmental samples with reference to the ICP-OES method.

Element	Unit	Sample	Found by AdSV Method	Found by ICP-OES Method	Δ% between the Two Methods	Ref.
Ce	µg g^−1^	Spinach Mushroom Rice Pu’er tea	0.428 2.885 0.545 0.338	0.436 2.905 0.552 0.341	1.9 0.7 1.3 0.9	[38]
La	µg g^−1^	Sand sample 1 Sand sample 2 Sand sample 3	1.06 ± 0.011 1.27 ± 0.024 1.47 ± 0.026	1.43 ± 0.015 1.52 ± 0.027 1.88 ± 0.031	34.9 19.7 27.9	[50]
Ce	µg g^−1^	Phosphate Sample 1 Phosphate Sample 2	1785 ± 65 1470 ± 54	1864 ± 57 1461 ± 43	4.4 0.6	[33]
Ce	M	Catalyst sample 1 Catalyst sample 2	1.59 × 10^−5^ 4.99 × 10^−5^	1.50 × 10^−5^ 5.0 × 10^−5^	5.7 0.2	[36]
Ce	M	Waste water Ce(III)-polluted water	5.88 × 10^−8^ 3.62 × 10^−8^	5.64 × 10^−8^ 3.55 × 10^−8^	4.1 1.9	[33]

## Data Availability

Not applicable.

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
