# Peer review of "Voltammetry in Determination of Trace Amounts of Lanthanides—A Review"

_molecules, 2023, doi:10.3390/molecules28237755_

Round 1
Reviewer 1 Report
Comments and Suggestions for Authors
The manuscript is a review on the determination of lanthanides by stripping voltammetry.
The introduction summarizes the main chemical properties of these elements, their sources, uses and environmental risks and the most popular analytical techniques used for their determination and an overview of stripping voltammetry and some hint on its application for lanthanide determination
The subsequent chapters describe the approaches proposed in the scientific literature for the voltammetric determination of these elements: the technique adopted (AdSV, CSV), the reagents added (mainly ligands), the kind of working electrode (e.g. mercury drop, noble metal or carbon electrodes) and the most popular electrode modifiers (e,g, metal film, carbon nanotubes, molecularly imprinted polymers). The main features of the methods is discussed. Much attention is paid to inorganic and organic potential interferents. Finally, the applications of the methods to a suite of matrices is described.
In my opinion, the manuscript is well written and the analytical procedures of interest are properly summarized and described. and an extensive body of literature is cited. I appreciated that the aspects of a voltammetric analysis were addressed in separate chapters, which made it easy to understand the content and to compare different methodologies.
The topic is surely up-to-date, since (as authors remark too) “the growing demand for these elements has resulted in these metals being included in the group of 20 critical mineral raw materials for the EU economy.
However, I noticed some inaccuracies that should be amended. Also the section on interferences should be revised.
For these reasons, I think the manuscript requires major revisions.
In particular:
- lines 74-77:In the sentence : “spectroscopic techniques such as graphite furnace atomic absorption spectrometry (GFAAS) [13], neutron activation analysis (NAA) [14], inductively coupled plasma optical emission spectrometry (ICP-OES) 76 [15-17], inductively coupled plasma - mass spectroscopy (ICP-MS) [18-20], X-ray fluores-77 cence spectroscopy (XRF) [21,22] and high-performance liquid chromatography (HPLC) 78 [23”
the list of analytical techniques is somewhat confused. Actually GFAAS, ICP-OES and XRF are SPECTROSCOPIC techniques; ICP-MS is a SPECTROMETRIC technique (because it does not rely on electromagnetic radiations; NAA and HPLC are neither spectroscopic nor spectrometric techniques.
I suggest to mention the technique in the order: spectroscopic techniques; spectrometric techniques (i.e. ICP-MS); other techniques.
- lines 79-81: the major disadvantages of NAA are: the need of a source of neutrons; the irradiated samples become radioactive. Therefore strict safety precautions must be adopted to avoid risks for the operators and for the environment.
- lines 76 and 81, page 2, and line 495, page 19. The abbreviations “ICP-AES” and “ICP-OES” are treated as if they referred to two different analytical techniques. Anyway, both abbreviations refer to the same instrument, working with the same principle. Nowadays, the abbreviation “ICP-OES” is more commonly used than “ICP-AES”, which was mainly used in the past.
So the sentences dealing with ICP-OES and ICP-AES should be merged and Table 4 should be amended.
- line 87: I do not agree with the statement that the performance of ICP-MS for lanthanide determination is “poor”. ICP-MS has been successfully used for the determination of trace levels of lanthanides in waters, soils and particulate matter. The interferences mentioned (intricacy of the emission spectrum as well as the amount of interference 88 from the major elements) refer to XRFS. There are no emission spectra in ICP-MS, which relies on mass spectra!
- line 90: the performance of HPLC depends on the detector used. In many cases, hyphenated techniques such as HPLC-ICP-MS are used. The authors should better explain which HPLC-based techniques are used for lanthanides
- line 95: spectrophotometry is not an electrochemical technique. I do not know if spectrophotometry can be applied to lanthanides determination, but in any case they would have low sensitivity.
- line 97: I agree that electrochemical techniques have “ease of handling, great sensitivity, quick response as well as cost efficacy”, but it is not possible to claim that they have a high selectivity, since they suffer from many interference effects. This is also evident from chapter 3 of the review.
- Table 1. The peak potentials of the analytes of interest should be reported.
- Table 1. The list of the potential interfering agents tested is not informative enough, since it does not explain which interferent actually has an impact on the lanthanide peak(s).
The following distinctions should be made; which elements (other than lanthanides) where found to interfere; which elements (other than lanthanides) did not interfere; which lanthanides where found to interfere; which lanthanides did not interfere. It should also be specified if some of the methods did not report data on the interference of other lanthanides.
- Page 9, lines 52-55: I do not understand if the method enabling to distinguish between heavy and light REEs gives rise a single peak for all light elements and another one for all heavy elements, or if each lanthanide gives rise to its own peak. The signals obtained with the method should be more clearly explained. On the other hand, I appreciated that the behaviour of Lu is well explained in lines 102-104
- Page 10, line 154: not all carbon electrodes have a renewable surface. This is valid only for carbon paste electrodes. For the other carbon electrodes, as well as the noble metal-based ones, the surface can be renewed only after time-consuming mechanical and electrochemical treatments.
- Chapter 3.1. The methods in which the interference of other lanthanides was not tested should be more clearly indicated. For instance, I appreciate that on page 15, line 380, it is clearly explained that only the interference of Ho was tested.
- Line 426::not only UV irradiation, but also mineralization, can be use to remove organic substances from the sampl
- Page 19, Line 504-508: too many data are reported for the phosphate concentrates, Their composition is not necessary,
- Tables 3 and 4: both tables report applications on environmental samples. The reason why these applications are divided into two tables, should be explained, or in alternative all environmental samples should be collected in the same table.
- page 20, lines 539-543: some more concepts on the interferences from other metals and/or lanthanides should be reported in the conclusions.
Reviewer 2 Report
Comments and Suggestions for Authors
The author described the method for determining rare earth elements in various samples using voltammetry, reviewed important aspects that affect the sensitivity of the developed method, and analyzed the voltammetric response of different rare earth elements on different materials. Overall, this study is well-written, but some content still needs to be slightly modified.
1. The English language needs to check carefully in the revision stage because of many careless mistakes in many positions.
2. Please the author divides some chapters into subheadings to make it more organized. For example, section 2.4.
3. Abstract: This section needs to be improved by presenting important findings.
4. Please change the topic ‘Conclusion’ to ‘Conclusions’.
5. The references are relative old. Please add more references which were published from 2019 till 2023.
Comments on the Quality of English LanguageThe English language needs to check carefully in the revision stage because of many careless mistakes in many positions.
Round 2
Reviewer 1 Report
Comments and Suggestions for Authors
The Authors have correctly and exhaustively answered about the questions I exposed. The English style has been improved. In my opinion, now the manuscript can be published.
